

# Evaluation of the validity of the Psychology Experiment Building Language tests of vigilance, auditory memory, and decision making

Brian Piper[1,2,3], Shane T. Mueller[4], Sara Talebzadeh[5] and Min Jung Ki[3]

[1] Neuroscience Program, Bowdoin College, Brunswick, ME, United States
[2] Department of Psychology, Willamette University, Salem, OR, United States
[3] School of Pharmacy, Husson University, Bangor, ME, United States
[4] Cognitive and Learning Sciences, Michigan Technological University, Houghton, MI, United States
[5] Department of Biology, Husson University, Bangor, ME, United States

## ABSTRACT

**Background.** The Psychology Experimental Building Language (PEBL) test battery (http://pebl.sourceforge.net/) is a popular application for neurobehavioral investigations. This study evaluated the correspondence between the PEBL and the non-PEBL versions of four executive function tests.

**Methods.** In one cohort, young-adults ($N = 44$) completed both the Conner's Continuous Performance Test ($_C$CPT) and the PEBL CPT ($_P$CPT) with the order counter-balanced. In a second cohort, participants ($N = 47$) completed a non-computerized (Wechsler) and a computerized (PEBL) Digit Span ($_W$DS or $_P$DS) both Forward and Backward. Participants also completed the Psychological Assessment Resources or the PEBL versions of the Iowa Gambling Task ($_{PAR}$IGT or $_{PEBL}$IGT).

**Results.** The between-test correlations were moderately high (reaction time $r = 0.78$, omission errors $r = 0.65$, commission errors $r = 0.66$) on the CPT. DS Forward was significantly greater than DS Backward on the $_W$DS ($p < .0005$) and the $_P$DS ($p < .0005$). The total $_W$DS score was moderately correlated with the $_P$DS ($r = 0.56$). The $_{PAR}$IGT and the $_{PEBL}$IGTs showed a very similar pattern for response times across blocks, development of preference for Advantageous over Disadvantageous Decks, and Deck selections. However, the amount of money earned (score–loan) was significantly higher in the $_{PEBL}$IGT during the last Block.

**Conclusions.** These findings are broadly supportive of the criterion validity of the PEBL measures of sustained attention, short-term memory, and decision making. Select differences between workalike versions of the same test highlight how detailed aspects of implementation may have more important consequences for computerized testing than has been previously acknowledged.

Corresponding author
Brian Piper, bpiper@bowdoin.edu, psy391@gmail.com

## INTRODUCTION

An increasingly large collection (>100) of classic and novel clinical psychology and behavioral neurology tests have been computerized and made freely available (http://pebl.sf.net) over the past decade. The latest version of Psychology Experiment Building

Language (PEBL) test battery (*Mueller, 2015*; *Mueller & Piper, 2014*; *Piper et al., 2015a*) was downloaded more than 21,000 times in 2015 and use continues to increase (*Fox et al., 2013*; *Lipnicki et al., 2009a*; *Lipnicki et al., 2009b*; *Piper, 2010*). The PEBL tests have been employed in studies of traumatic brain injury (*Danckert et al., 2012*), behavioral pharmacology (*Aggarwal et al., 2011*; *Lyvers & Tobias-Webb, 2010*), aging (*Clark & Kar, 2011*; *Piper et al., 2012*), Parkinson's disease (*Peterson et al., 2015*) and behavioral genetics (*Wardle et al., 2013*; *González-Giraldo et al., 2014*) by investigators in developed and developing countries, and the tests have been administered in many languages. A key step in PEBL battery development is to evaluate criterion validity (i.e., the extent to which its dependent measures predict other existing measures) by determining whether performance on PEBL tests is similar to the established versions of the tests. Although the PEBL tests were developed based on the methods sections of the peer reviewed literature, this direct comparison is important because some potentially important procedural details may have been omitted, described ambiguously, or misinterpreted.

Four tests were selected for the present report for comparison between the PEBL and non-PEBL (i.e., established) versions: the Continuous Performance Test (CPT), Digit Span Forward (DS-F), DS Backward (DS-B), and the Iowa Gambling Task (IGT). These tests were chosen because they assess theoretically important constructs (vigilance, attentional capacity, short-term memory, and decision making), have an extensive history, and their neural substrates have been examined in lesion and neuroimaging studies. Each of these measures is described in more detail below.

## Continuous Performance Test (CPT)

CPTs have an extensive history and exist in multiple forms (*Mackworth, 1948*; *Rosvold et al., 1956*; *Anderson et al., 1969*; *Earle-Boyer et al., 1991*; *Greenberg & Waldman, 1993*; *Dougherty, Marsh & Mathias, 2002*; *Riccio et al., 2002*). These tests require participants to maintain vigilance and respond to the presence of a specific stimulus within a set of continuously presented distracters. A key quality of a CPT is that, rather than being a series of trials that each require a response; a CPT is presented as a continuous series of stimuli whose timing does not appear to depend on the speed or presence of a response, and so it represents a continuous mental workload that has been used to assess vigilance, alertness, attention, and related concepts. The CPT, version II, of Conners and colleagues (hence-forth $_C$CPT) has been widely used as a neuropsychological instrument to measure attention in children and adults (*Conners, 2004*; *Piper et al., 2010*; *Piper et al., 2011*). The fourteen minute $_C$CPT involves responding to target letters (letters A–S presented for 1, 2, or 4 sec each) and inhibiting responses to foils (the letter X). Dependent measures include response times (RT), the variability of RT, the absence of response to target stimuli (omission errors), and responses to the foil (commission errors). There is some debate regarding the utility of the $_C$CPT to aid in a diagnosis of Attention Deficit Hyperactivity Disorder (ADHD) (*Cohen & Shapiro, 2007*; *McGee, Clark & Symons, 2000*). Overall, the strengths of this instrument are its objectivity, simplicity, brevity, a sizable normative sample (*Conners & Jeff, 1999*; *Homack & Riccio, 2006*), and it has been shown to be sensitive to psychostimulants used to treat attention disorders (*Solanto et al., 2009*). In
addition, the neural substrates of vigilance have been characterized and involve a network that includes the prefrontal, frontal, and parietal cortex and the striatum (*Ogg et al., 2008*; *Riccio et al., 2002*).

### Digit Span Forward and Backward (DS-F and DS-B)

DS type tests are found in the Wechsler assessments as well as in other neuropsychological batteries. A string of numbers is presented (e.g., 7, 1, 6 at a rate of one digit per second) and the participant either repeats them in the same (DS-F) or the reverse (DS-B) sequence. Although DS-F and DS-B are procedurally similar, and they are sometimes both viewed as simple short-term memory tasks (*St Clair-Thompson & Allen, 2013*), the former is sometimes treated as a measure of "pure storage" whereas the latter is viewed as involving more executive control and thus considered a "working memory" task (*Lezak et al., 2012*). DS-B induces greater activity in the prefrontal cortex than DS-F (*Keneko et al., 2011*).

Previously, a direct comparison of DS by mode of administration revealed lower DS Forward and Backward when completed over the telephone with voice recognition as compared to in-person administration (*Miller et al., 2013*). However, a moderate correlation ($r = .53$) in DS total was identified with traditional and computerized administration (*Paul et al., 2005*).

### Iowa Gambling Test (IGT)

Bechara and colleagues at the University of Iowa College of Medicine developed a novel task to quantify abnormalities in decision making abilities. Originally, what became known as the Iowa Gambling Task (IGT) involved selecting cards from four physical decks of cards. Each deck had a different probability of wins versus losses. Two decks are Disadvantageous and two are Advantageous, because some deck selections will lead to losses over the long run, and others will lead to gains. Neurologically intact participants were reported to make the majority (70%) of one-hundred selections from the Advantageous (C & D) decks. In contrast, patients with lesions of the prefrontal cortex showed the reverse pattern with a strong preference for the Disadvantageous (A & B) decks (*Bechara et al., 1994*, although see *Buelow & Suhr, 2009*; *Steingroever et al., 2013*). However, another research team, employing a gambling task that they programmed, determined that college-aged adults showed a response pattern that is very similar to patients with frontal lesions (*Caroselli et al., 2006*). Similarly, the median response among a moderate-sized sample ($N = 39$) of college students from the southwestern United States was to make more selections from Disadvantageous than Advantageous Decks on the Psychological Assessment Resources (PAR) version of the IGT (*Piper et al., 2015b*). IGT type tasks have become increasingly popular for research purposes to examine individual differences in decision making including in pathological gamblers, substance abusers, ADHD, and in other neurobehavioral disorders (*Buelow & Suhr, 2009*; *Verdejo-Garcia et al., 2007*). One key characteristic of the IGT is that there is substantial carryover of learning with repeated administrations in normal participants (*Bechara, Damasio & Damasio, 2000*; *Bull, Tippett & Addis, 2015*; *Fernie & Tunney, 2006*; *Piper et al., 2015a*; *Verdejo-Garcia et al., 2007*). *Bechara (2007)*, in conjunction with PAR, distributes a computerized version of the IGT.

The IGT is also one of the more widely employed tests in the PEBL battery (*Bull, Tippett & Addis, 2015*; *Hawthorne, Weatherford & Tochkov, 2011*; *Lipnicki et al., 2009a*; *Lipnicki et al., 2009b*; *Mueller & Piper, 2014*) and so itself has been used in many different contexts. Many variations on IGT procedures have been developed over the past two decades. The $_{PEBL}$IGT employs consistent rewards and punishment (e.g., −\$1,250 for each selection from Deck B) as described by *Bechara et al., (1994)*. The $_{PAR}$IGT utilizes the ascending schedule of rewards and punishments (e.g., −\$1,250 for early deck selections and decreasing by \$250 increments) (*Bechara, Tranel & Damasio, 2000*).

The primary objective of this report was to determine the similarity between the PEBL and non-PEBL versions of these executive function measures. A common strategy to test development would be to administer both the PEBL and non-PEBL versions to tests to participants with the order counter-balanced. Interestingly, a prior study administered the PEBL digit span forward, a continuous performance test with some procedural similarities to the CPT, and the IGT to young-adults twice with a two-week inter-test interval and identified moderate to high test-retest correlations on measures of attention (Spearman rho = .69–.72) and digit-span (rho = .62) while the total money earned on the IGT was less consistent (rho = .22) (*Piper et al., 2015a*). These findings suggest that the approach of administering both PEBL and non-PEBL versions and examining correlations across platforms might be viable for DS and the CPT but not the IGT. The IGT dataset was also used to critically examine the sensitivity of the IGT to identify clinically meaningful individual differences in decision making abilities. The commercial distributors of an IGT purport that neurologically intact and those that have suffered a brain insult should score quite differently. If "normal" college students completing the IGT showed a pattern of responding that would be labeled impaired (as has been shown earlier; cf. *Caroselli et al., 2006*; *Piper et al., 2015b*), these findings would challenge the construct validity of this measure. Consequently, participants in this study completed PEBL and/or non-PEBL versions of the same tests. Correlations across platforms were determined for the CPT and DS and the pattern of responses were evaluated for each IGT. Where applicable, intra-test correlations were also examined as this is one criteria used to evaluate test equivalence (*Bartram, 1994*).

## MATERIALS & METHODS

### Participants

The participants ($N = 44$; Age = 18–24, Mean = 18.7 ± 0.2; 68.2% female; 23.9% non-white; 7.3% ADHD) were college students receiving course credit in the CPT study. A separate cohort (N = 47; Age = 18–34, Mean = 18.8 ± 0.3; 59.6% female; 14.9% non-white; 10.6% ADHD) of college students completed the DS/IGT study and also received course credit.

### Procedures

All procedures were approved by the Institutional Review Board of Willamette University (first cohort) or the University of Maine, Orono (second cohort). Participants were tested individually with an experimenter in the same room. Each participant completed an

informed consent and a short demographic form which included items about sex, age, and whether they had been diagnosed by a medical professional with ADHD. Next, the first cohort completed either the $_P$CPT or Version II of the $_C$CPT, including the two-minute practice trial, with the order counter-balanced on desktop computers running Windows XP and not connected to the internet. As data collection for each CPT takes 14 min and is intentionally monotonous, the PEBL Tower of London (*Piper et al., 2012*) was completed between each CPT as a brief ($\approx$ 5 min) distractor task. The $_P$CPT was modified from the default in PEBL version 0.11 such that a mid-test break was removed and the instructions were analogous to the $_C$CPT. The instructions of the $_P$CPT were:

> You are about to take part in an experiment that involves paying attention to letters on a screen. It will take about 14 min. You will see letters presented on a screen quickly. Your goal is to press the spacebar as fast as possible after each letter, except if the letter is an 'X'. DO NOT RESPOND to X stimuli.

A total of 324 target letters (A, B, C, D, E, F, G, H, I, J, K, L, M, O, P, Q, R, S, U) and 36 foils (X) were presented with an inter-stimulus interval of 1, 2, or 4 s. The primary dependent measures were the RT on correct trials in ms, the standard deviation (SD) of RT, omission and commission errors. The $_P$CPT source code is also at: https://github.com/stmueller/pebl-custom.

The second cohort completed a short demographic form (described above) followed by the PEBL and non-PEBL tasks (DS-F, DS-B, and IGT) with the order counterbalanced across testing sessions. PEBL, version 0.14, was installed on Dell laptops (Latitude E6410 and 6420) running Windows 7. Both laptops were connected to Dell touchscreen monitors (20" model number 0MFT4K) which were used for selecting responses on the IGT.

The Wechsler DS ($_W$DS) consists of two trials for each number of items each read aloud by the experimenter at a rate of one per second beginning with two items. Discontinuation occurred when both trials for a single number of items were answered incorrectly. The maximum total score for DS Forward and Backward is sixteen and fourteen, respectively. The PEBL Digit Span ($_P$DS) source code was modified slightly from the default version so that stimuli were presented via headphones (one per 1,000 ms) but not visually (PEBL script available at: https://github.com/stmueller/pebl-custom) in order to be more similar to the $_W$DS. Two trials were completed for each number of items starting with three items. Digit stimuli were generated randomly such that each sequence contained no more than one of each digit. Discontinuation occurred when both trials for a single number of items were answered incorrectly. An important methodological difference between the $_W$DS and the $_P$DS involves how responses are collected. The traditional $_W$DS involves oral responses coded by the experimenter. The $_P$DS involves typed input with the response sequence visible on-screen as it is made. Furthermore, blank entries are permitted and participants have the ability to delete erroneous responses (see Supplemental Information for the source code and task instructions).

The $_{PAR}$IGT (Version 1.00) was installed on a laptop (Dell Latitude E6410) with headphones. The administration instructions were shown and read/paraphrased for the participant (*Bechara, Damasio & Damasio, 2000*; *Bechara, 2007*) and the default settings

**Table 1 A comparison of the Bechara IGT distributed by Psychological Assessment Resources (PAR) and the Mueller and Bull IGT distributed with version 0.14 of the Psychology Experiment Building Language (PEBL).**

|  | PAR | PEBL |
|---|---|---|
| Instructions (words) | 441 | 379 |
| Visual post-trial feedback | yes | yes |
| Auditory post-trial feedback | yes | yes |
| Post-trial wait period | yes | yes |
| Deck A: Reward ($) | 80, 90, 100, 110, 120, 130, 140, 150, 160, 170 | 100 |
| Deck A: Punishment ($) | 150, 200, 250, 300, 350 | 150, 200, 300, 350 |
| Deck B: Reward ($) | 80, 90, 100, 110, 120, 130, 140, 150, 160, 170 | 100 |
| Deck B: Punishment ($) | 1,250, 1,500, 1,750, 2000, 2,250, 2500 | 1,250 |
| Deck C: Reward ($) | 40, 45, 50, 55, 60, 65, 70, 75, 80, 85, 90, 95 | 50 |
| Deck C: Punishment ($) | 25, 50, 75 | 25, 50, 75 |
| Deck D: Payoff ($) | 40, 45, 50, 55, 60, 65, 70, 75, 80, 85, 90, 95 | 50 |
| Deck D: Loss ($) | 250, 275, 300, 350, 275 | 250 |
| Trials | 100 | 100 |
| Cards/deck (maximum) | 60 | 100 |
| Standardized ($T_{50}$) scores | yes | no |
| Cost | $574[a] | $0 |

**Notes.**
[a] Price in U.S.D. on 3/5/2016.

were used. The $_{PEBL}$IGT was also administered with the order counterbalanced. Because others have identified pronounced practice effects with the IGT (*Bechara, Damasio & Damasio, 2000*; *Bull, Tippett & Addis, 2015*; *Birkett et al., 2015*; *Fontaine et al., 2015*; *Verdejo-Garcia et al., 2007*) and we found that the amount earned increase by 106.3% on the second administration (*Piper et al., 2015a*), only data from the IGT administered first was examined. The $_{PEBL}$IGT has modifications contributed by P. N. Bull (Supplemental Information at: https://github.com/stmueller/pebl-custom) and is a more refined version of the task than has been used previously (*Hawthorne, Weatherford & Tochkov, 2011*; *Lipnicki et al., 2009a*; *Lipnicki et al., 2009b*; *Piper et al., 2015b*). If scores go below zero, participants will receive a second $2,000 loan. Importantly, the $_{PEBL}$IGT is based on the procedures described in *Bechara et al. (1994)* while the $_{PAR}$IGT is based on those described in great detail in *Bechara, Tranel & Damasio (2000)*. The instructions are 14% shorter on the $_{PEBL}$IGT but perhaps the largest procedural difference is the negative consequences of Disadvantageous Decks are amplified in the $_{PAR}$IGT (Table 1).

## Statistical analyses

The overall data analytic strategy to evaluate test validity was tailored to the characteristics of each test. For the CPT and DS, this involved calculating intra-test correlations (*Bartram, 1994*), cross-test correlations ($r = .30–.70$ are moderate, $r > .70$ are high), and comparing means across platforms. Similar intra-test correlations, high and significant cross-test correlations, and small/non-significant differences in means are supportive of test similarity. Due to substantial practice effects on the IGT (*Bull, Tippett & Addis, 2015*; *Fontaine et al., 2015*; *Piper et al., 2015a*), and that not all of the second IGT tests were completed, due to

**Table 2** Age and sex corrected percentiles of the participants ($N = 44$) on the Conner's Continuous Performance Test.

|  | Min | Max | Mean | SEM |
|---|---|---|---|---|
| Reaction time | 1.0 | 94.2 | 18.6 | 2.9 |
| Reaction time SE | 1.0 | 99.0 | 44.3 | 5.0 |
| Omissions | 20.8 | 99.0 | 47.5 | 3.7 |
| Commissions | 19.0 | 99.0 | 74.4 | 3.7 |
| $d'$ | 10.9 | 97.3 | 69.6 | 3.3 |
| $B$ | 24.7 | 78.1 | 36.0 | 1.6 |

Notes.
SE, standard error.

participant time limitations, data from the second IGT was not examined and analyses instead focused on determining the response patterns within the first test and whether they were similar across platforms. The standardized (age and sex corrected) scores (percentiles) of the sample were reported for the $_C$CPT and $_{PAR}$IGT. The $_P$CPT output text files were imported into Excel and all analyses were subsequently conducted using Systat, version 13.0. The distribution on some measures (e.g., RT), were, as anticipated, non-normal, therefore both Pearson ($r_P$) and Spearman rho ($r_S$) correlation coefficients were completed as was done previously (*Piper et al., 2015a*). As the $_P$CPT default settings express the variability in RT slightly differently (SD) than the cCPT (SE), the PEBL output was converted to the SE according to the formula $SD/(N-1)^{0.5}$ where N is the total number of correct trials across the three inter-trial intervals. Differences in intra-test correlations (e.g., omission by commission errors) between the $_P$CPT and $_C$CPT were evaluated with a Fisher r to Z transformation (http://vassarstats.net/rdiff.html). The 95% Confidence Interval (CI) of select Pearson correlations was determined (http://vassarstats.net/rho.html) and the effect size of group differences was expressed in terms of Cohen's *d* (http://www.cognitiveflexibility.org/effectsize/) with 0.2, 0.5, and 0.8 interpreted as small, medium, and large. As the $_W$DS starts at an easier level (2 digits) than the $_P$DS (3 digits), two additional points were added to each (Forward and Backward) $_P$DS for comparison purposes. The primary dependent measure on the IGT was Deck selections but Response Times on each Block of twenty-trials and the compensation (score minus loan) for each trial was also documented. The NET was calculated as Advantageous minus Disadvantageous Deck selections. Mean data are presented with the standard error of the mean (SEM) and $p < .05$ considered statistically significant although statistics that met more conservative alpha levels (e.g., .0005) are noted.

## RESULTS

### Continuous Performance Test (CPT)

Substantial individual differences in sustained attention were observed in this sample. The percentiles ($\pm$SEM) for each $_C$CPT measure are shown in Table 2.

Mean reaction time on correct trials was slightly (by 12 ms) shorter on the PCPT, which was statistically significant ($_C$CPT $= 327.1 \pm 6.5$, Kurtosis $= 3.82$, $_P$CPT $= 315.2 \pm 4.7$,

**Table 3 Intra-test Continuous Performance Test Spearman correlations (Conners/PEBL).**

|  | **A.** | **B.** | **C.** |
|---|---|---|---|
| A. Reaction-Time (msec) | +1.00 |  |  |
| B. Reaction-Time SE | +0.54[a]/+0.18 | +1.00 |  |
| C. Omission Errors | +0.20/+0.03 | +0.53[a]/+0.35[a] | +1.00 |
| D. Commission Errors | −0.38[a]/−0.36[a] | +0.16/+0.29 | +0.32[a]/+0.36[a] |

**Notes.**
[a] $p < .05$.

Kurtosis $= 0.30$, $t(43) = 2.91, p < .01, d = .48$). The difference in the SE of RT was clearly different ($_C$CPT $= 5.3 \pm 0.4$, Kurtosis $= 6.22$, $_P$CPT $= 3.3 \pm 0.5$, Kurtosis $= 37.86$, $t(43) = 5.60, p < .0005, d = .87$) but there was no difference for omission errors ($_C$CPT $= 2.6 \pm 0.6$, Kurtosis $= 6.41$, $_P$CPT $= 2.3 \pm 0.7$, Kurtosis $= 26.00$, $t(43) = 0.51$, $p = .61$) or commission errors ($_C$CPT $= 18.1 \pm 1.1$, $_P$CPT $= 17.3 \pm 1.0$, $t(43) = 0.96$, $p = .34$).

The inter-test correlations were generally satisfactory. The correlation was excellent for reaction time ($r_P(42) = +.78$, 95% CI [.63–.87]; $r_S(42) = +.80, p < .0005$, Fig. 1A). The cross-platform association for reaction time variability was also moderate ($r_P(42) = +.66, p < .01$, 95% CI [.46–.80]; $r_S(42) = +.27, p = .076$) but this association should be viewed with caution as removal of one extreme score (15.9, Grub's test $= 4.18, p < .01$; 23.3, Grubs test $= 6.26, p < .01$) reduced this correlation considerably ($r_P(41) = +.20$, 95% CI [−.11–+.47], $p = .19$; Figure S1). Omission errors ($r_P(42) = +.65$, 95% CI [.44–.79], $p < .0005$, $r_S(42) = +.31, p < .05$) and commission errors ($r_P(42) = +.66$, 95% CI [.45–.80], $r_S(42) = +.66, p < .0005$) showed good correlations across tests (Figs. 1B and 1C).

An analysis of the intra-test Spearman correlations among the variables of each test was also conducted (Table 3). Several significant correlations were identified. However, with the exception of a trend for the RT SE ($p = .055$), the correlations did not differ across tests.

## Digit Span (DS)

Figure 2A shows the anticipated higher score for Forward ($10.0 \pm 0.3$, Min $= 6$, Max $= 13$) relative to Backward ($6.3 \pm 0.3$, Min $= 3$, Max $= 11$) on the $_W$DS. The correlation between Forward and Backward was moderate ($r_P(45) = .43$, 95% CI [.16–.64], $p < .005$; $r_S(45) = .41, p < .005$).

Figure 2A also depicts an elevated score for Forward ($10.5 \pm 0.4$, Min $= 3$, Max $= 15$) compared to Backward ($8.2 \pm 0.3$, Min $= 4$, Max $= 12$, $t(46) = 5.10, p < .0005$) for the $_P$DS. The correlation between Forward and Backward was not significant ($r_P(45) = .22$, 95% CI [−.07–.48], $p > .10$; $r_S(45) = .28, p = .054$). The $_P$DS − B was significantly higher than $_W$DS − B ($t(46) = 6.43, p < .0005$), which is likely to stem from using a visual/manual response entry rather than the verbal mode used in the $_W$DS − B.

The correlation between computerized and non-computerized DS was intermediate for Forward ($r_P(45) = .42$, 95% CI [.15–.63], $p < .005$; $r_S(45) = .45, p < .005$) and Backward

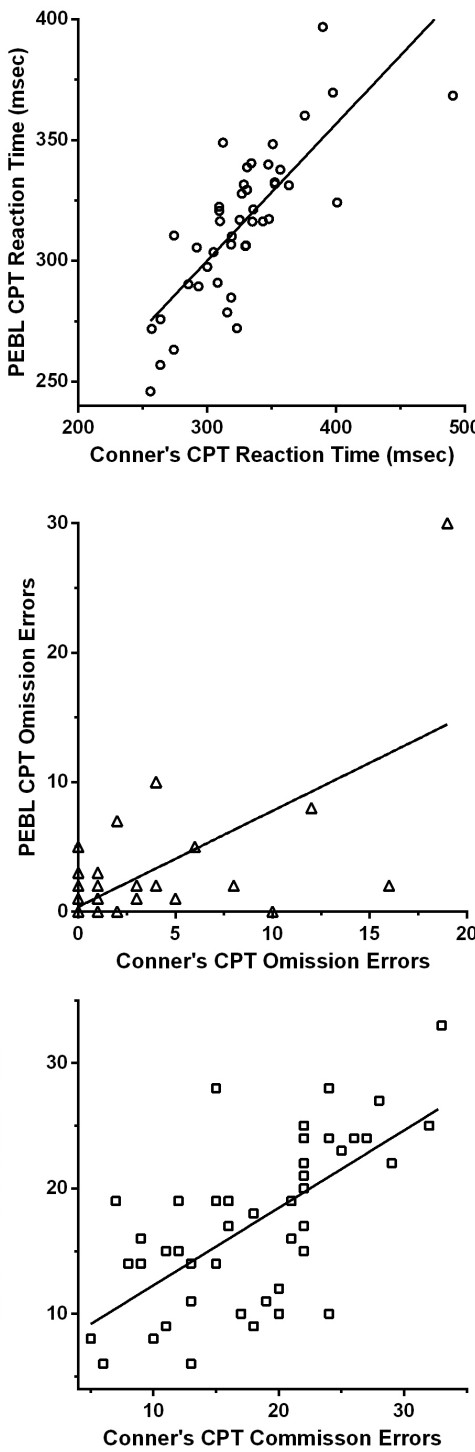

**Figure 1** Scatterplots depicting the association between measures on the Psychology Experiment Building Language and the Conner's Continuous Performance Test including reaction time (top: $r(42) = +.78$, 95% CI [.63–.87], $p < .0005$), omission errors (middle: $r_P(42) = +.65$, 95% CI [.44–.79], $p < .0005$) and commission errors (bottom: $r(42) = +.66$, 95% CI [.45–.80], $p < .0005$).

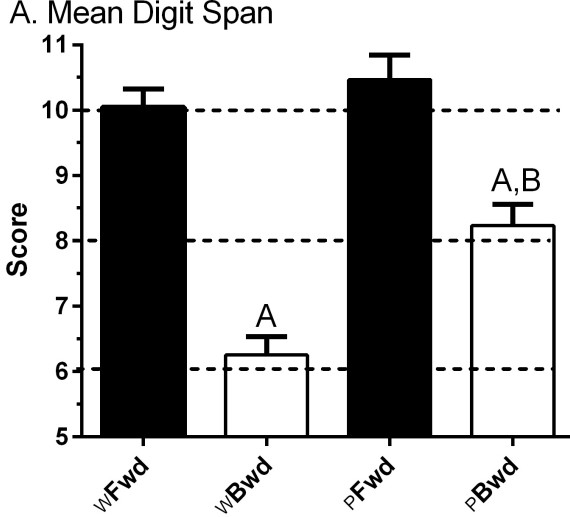

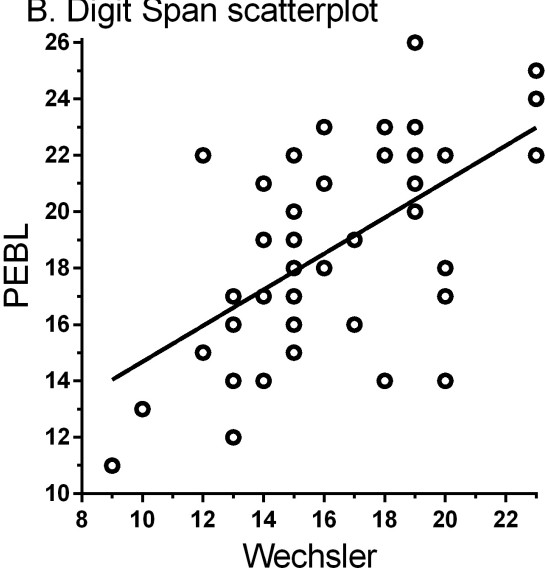

**Figure 2** (A) Wechsler (W) and Psychology Experiment Building Language (PEBL) Digit Span Forward (Fwd) and Backward (Bwd). [A]$p < .0005$ versus Digit Span Forward, [B]$p < .0005$ versus PEBL Digit Span Forward. (B) Scatterplot of Wechsler by PEBL Digit Span total ($r_P(45) = .56$, 95% CI [.31–.74], $p < .0005$).

($r_P(45) = .49$, 95% CI [.24–.68], $p < .001$; $r_S(45) = .467, p < .001$). Figure. 2B shows the association between the DS total (Forward + Backward) across test modalities was moderate ($r_S(47) = .51, p < .0005$).

## Iowa Gambling Task (IGT)

Data-analysis was completed by examining each test separately and then comparing across platforms. The NET 1–5 percentile score was $38.0 \pm 4.4$ (Min = 5, Max = 90) on the PARIGT. The standardized ($T_{50}$) score was $47.2 \pm 1.5$ (Min = 34.0, Max = 63.0) which was non-significantly lower than the normative mean of 50 (one sample

$t(23) = 1.91, p = .069$). A repeated measures ANOVA on Response Times revealed a main effect of Block ($F(1.81, 41.69) = 21.10, p < .0005$). Response Times showed a clear decrease over the course of the session with shorter times on Block 2 ($t(23) = 4.49, p < .0005$), Block 3 ($t(23) = 5.93, p < .0005$), Block 4 ($t(23) = 5.42, p < .0005$) and Block 5 ($t(23) = 5.07, p < .0005$) relative to Block 1 (Fig. 3A). Responses on the first Block showed a trend favoring Disadvantageous over Advantageous Decks ($t(23) = 1.90, p = .07$) with the reverse pattern on the last Block (Fig. 3C). Similarly, there was a trend toward greater Advantageous selections on Block 5 ($11.0 \pm 0.9$) compared to Block 1 ($t(23) = 1.83, p = .081$). Across all Blocks, participants made fewer selections from Deck A' compared to Deck B' ($t(23) = 8.98, p < .0005$), Deck C' ($t(23) = 3.48, p \leq .002$) or Deck D' ($t(23) = 3.65, p \leq .001$). Participants made more selections from Deck B' compared to Deck C' ($t(23) = 2.79, p \leq .01$) or Deck D' ($t(23) = 2.72, p < .02$, Fig. 3E). Almost half (45.8%) of participants made more selections from Disadvantageous (C' + D') than Advantageous (C' + D') Decks. Figure 4A shows the Deck selections on each trial for a participant with the median NET 1–5. Half (50.0%) of participants received the second $2,000 loan. The amount earned (score minus loan) increased during the Block 1, dropped below zero during Block 3, and was negative by test completion ($-\$1,099.58 \pm 191.20$, Min $= -3,015$, Max $= 1,475$, Fig. 3G).

A repeated measures ANOVA on Response Times revealed a main effect of Block ($F(2.07, 37.17) = 12.27, p < .0005$) on the $_{PEBL}$IGT. Relative to the first Block, RTs were significantly shorter on Block 2 ($t(18) = 2.85, p < .02$), Block 3 ($t(18) = 7.45, p < .0005$), Block 4 ($t(18) = 4.26, p \leq .0005$), and Block 5 ($t(16) = 4.59, p < .0005$, Fig. 3B). Across all five Blocks, RTs were equivalent on the $_{PEBL}$IGT ($668.4 + 118.0$) and $_{PAR}$IGT ($786.4 \pm 49.1, t(24.2) = .92, p = .37$). There were more selections from the Disadvantageous than the Advantageous Decks on Block 1 ($t(18) = 2.98, p < .01$, Fig. 3D). When collapsing across the five Blocks, over-two thirds (68.4%) of respondents made more selections from Disadvantageous than Advantageous Decks. Fewer selections were made from Deck A compared to Deck B ($t(18) = 4.27, p < .0005$) or Deck D ($t(18) = 2.45, p < .03$). There was a trend towards more selections on Deck B compared to Deck C ($t(18) = 2.05, p = .055$, Fig. 3F). Figure 4B depicts the Deck selections over the course of the test for a participant with the median NET 1–5. Very few (10.5%) participants received the second $2,000 loan. Compensation, defined as the score minus the loan, grew during the Block 1, dropped towards zero in Block 2, and stayed negative for the remainder of the test. A comparison of compensation across platforms ($t$-test) revealed that the $_{PEBL}$IGT money was significantly lower than $_{PAR}$IGT during trials 16 to 18 and 23 but higher from trial 74 until test completion ($-\$269.74 \pm 255.93$, Min $= 2,425$, Max $= 1,950$, Fig. 3G).

## DISCUSSION

The PEBL software is becoming a widely-used tool in the social and biomedical sciences (*Mueller & Piper, 2014*). Although this widespread use in numerous contexts has helped to establish the general reliability and validity of specific tests, the publication of additional

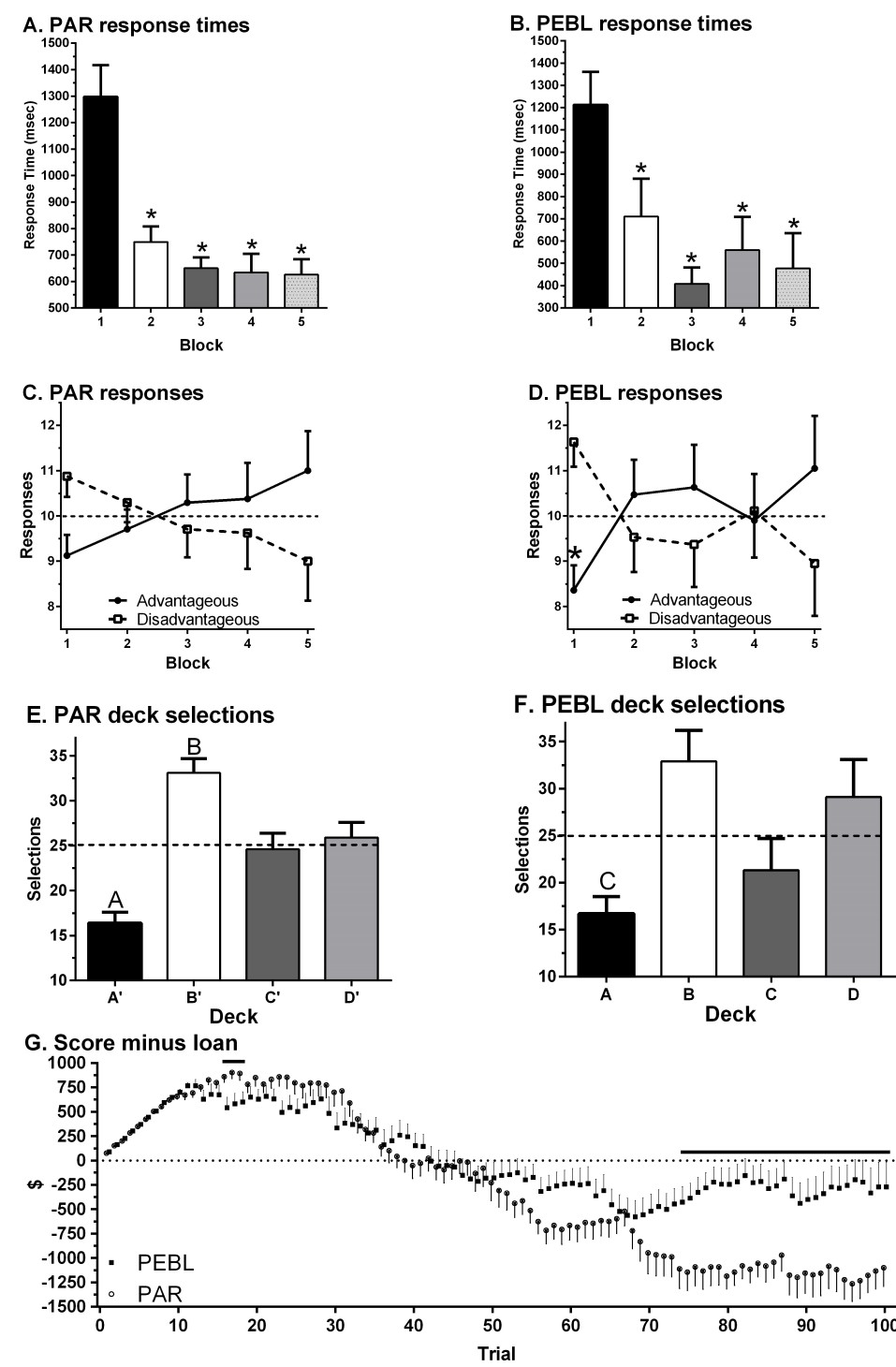

**Figure 3** Response times on the Psychological Assessment Resources (PAR, A) and Psychology Experiment Building Language (PEBL, B) Iowa Gambling Task by block of 20 trials (*$p < .0005$). Selection of advantageous and disadvantageous decks (C, D) (*$p < .05$ versus disadvantageous on block 1). Selection of each deck (E, F) ([A]$p < .005$ versus Deck B, C, or D; [B]$p < .05$ versus Deck C and D; [C]$p < .05$ versus Deck B). Compensation by trial (G) (horizontal line indicates $p < .05$).

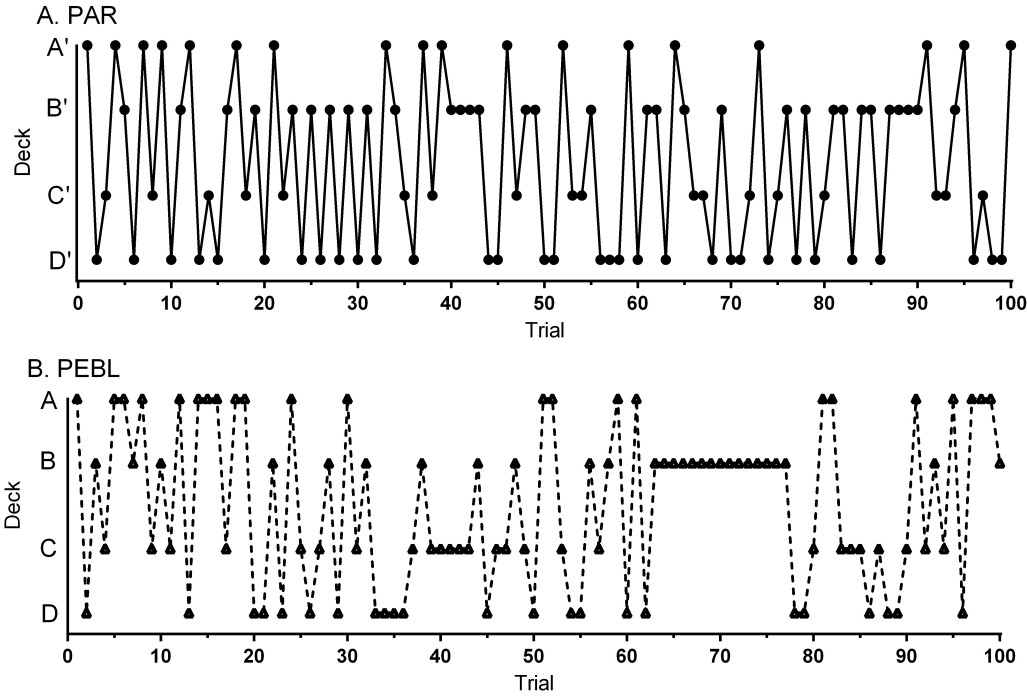

**Figure 4** Deck selections over one-hundred trials for the participant (a 34 year-old, Native American female) with the median NET1–5 (0) on the Psychological Assessment Resources (PAR) Iowa Gambling Task (A). Deck selections for the participant (a 18 year-old Native American male) with the median NET1–5 (−2) on the Psychology Experiment Building Language (PEBL) Iowa Gambling Task (B).

systematic validation studies comparing their results to existing tests will help establish their suitability for use in basic research and clinical neuroscience applications, including assessment. This report identifies some procedural similarities, and also differences, between the PEBL and commercial versions of ostensibly equivalent tests.

## CPT Tests

The CPT developed by Conners and colleagues has been, and will likely continue to be, an important instrument widely employed for applied and research purposes. The mean RT, variability of RT, omission and commission errors are similar to those reported previously with college students as participants (*Burton et al., 2010*). Moderate to strong correlations across tests were observed on the CPT measures across platforms. The origin of any inter-test differences is multifaceted and could include procedural details (e.g., software algorithms), interactions between software and hardware, particularly for RTs (*Plant & Quinlan, 2013*), or participant variance due to repeated testing. Importantly, the inter-test reliability of the pCPT and the cCPT are bound by the test-retest reliability of both measures. Previous research has established moderate to high test-retest reliability for the cCPT, in the same range as our inter-test reliability measures. For example, *Conners (2004)* reported test-retest correlations of 0.55–0.84 when the cCPT was administered twice with an inter-test interval of two weeks. Similarly, in a study of twelve children taking the cCPT, *Soreni et al. (2009)* found the inter-class correlation (ICC) coefficients

for ommission errors: .09; commission errors: .72; RT: .76; and RTSE of .63. In a similar study with 39 children aged 6–18 over a 3–8 month interval, *Zabel et al. (2009)* found ICC of .39 and .57 for omission and commission errors, .65 for hit RTs, and .48 for RT variability, concluding that there was substantial variability in these measures even for their large age range. Using a similar go/no-go CPT, *Kuntsi et al. (2005)*, showed for a group of 47 children, inter-class r scores ranged from .7–.88 on RT scores; 0.26–.83 on SD of RT, and .54–.7 on commission errors. Thus, the between-test correlations in our homogeneous sample of college students compared favorably to previously-reported test-retest correspondence scores on CPT tests. Although the experience of the participants was similar when completing the $_C$CPT and the $_P$CPT, some of the algorithms employed in the $_C$CPT are unpublished or could not be verified by the authors. This is particularly a concern for the signal detection measures (*Stanislaw & Todorov, 1999*) and therefore $d'$ and *Beta* were not compared across platforms. Notably, similarity of intra-test correlations is one criterion for the equivalence of measures (*Bartram, 1994*). The pattern of results with this sample identified in Table 2 generally supports this criterion for the $_P$CPT.

## DS-F and DS-B Tests

DS type tasks have an extensive history and have been implemented in an analogous format to the $_W$DS for over a century (*Richardson, 2007*). Importantly, the test-retest reliability of $_W$DS is moderate ($r = .68$) (*Dikmen et al., 1999*). DS-F did not differ between $_W$DS and $_P$DS. Although DS-B was less than DS-F for the $_W$DS and the $_P$DS, the magnitude of reduction was attenuated on the $_P$DS. A subset of participants ($\approx$15%) either were rehearsing the digits aloud or on the keyboard while they were being presented on the $_P$DS. Use of these strategies could change the fundamental nature of the constructs being measured. It is important to emphasize that although stimuli are present aurally for both the $_W$DS and the $_P$DS, response execution is oral for the $_W$DS but typed for the $_P$DS. The format of how stimuli is presented and executed is known to produce detectable differences (*Karakas et al., 2002*). The correlation between the $_P$DS and the $_W$DS was only moderate. This could be due to modality effects or the use of a college-aged sample may have resulted in a restriction of range which attenuated the associations. In principle, voice recognition algorithms would make $_W$DS and $_P$DS more similar, and an alternative to self-administration is to have a researcher or clinician enter the responses for the study participant, so that he or she must respond vocally. Other investigators that are refining this technology have identified moderate correlations across modalities (Forward = .48, Backward = .50) but difficulties recognizing the responses of participants with accents is not trivial (*Miller et al., 2013*). More generally, perhaps the notion of the $_W$DS as the "gold standard" is questionable. Computerized administration offers the potential of delivering stimuli at a more consistent rate, intensity, and clarity than traditional methods (*Woods et al., 2011*). The use of more trials per number of digits and alternative procedures for advancement to the difficulty threshold may improve the precision of DS measurement.

## IGT tests

The IGT is sometimes described as a "one-shot" measure of executive function. Several laboratories have identified significant practice effects on the IGT (*Bechara, Damasio*

& Damasio, 2000; Bull, Tippett & Addis, 2015; Fernie & Tumney, 2006; Fontaine et al., 2015; Piper et al., 2015a; Verdejo-Garcia et al., 2007). As such, the primary goal of this investigation was not to attempt to evaluate correlations between the $_{PEBL}$IGT and the $_{PAR}$IGT and instead examined response patterns within each test. The $_{PEBL}$IGT and the $_{PAR}$IGT have many procedural similarities but also some differences (Table 1) which may not be widely appreciated. Although there were pronounced individual differences, the $_{PAR}$CPT percentiles were well different than fifty for this collegiate sample. On the primary dependent measure (deck selections), there was a high degree of similarity between the $_{PAR}$IGT and $_{PEBL}$IGT. For example, the development across trials for a preference of Advantageous over Disadvantageous Decks was evident with both tests (Figs. 2C and 2D). The choice of individual decks (e.g., Deck B was twice as commonly selected as Deck A) was identified with the $_{PAR}$IGT and the $_{PEBL}$IGT (Figs. 2E and 2F). Response times across blocks were virtually identical in both computerized platforms (Figs. 2A and 2B). However, the compensation awarded at the end of the test, a secondary measure (Bechara, 2007), was examined to identify any impact of the procedural differences in Table 1. Overall, compensation was significantly greater on the $_{PEBL}$IGT. The losses associated with Disadvantageous Decks in the $_{PEBL}$IGT (Deck B = −$1,250) are much less pronounced than those in the $_{PAR}$IGT punishments (Deck B starts at −$1,250 but increases up to −$2,500). Although this procedural difference did not produce other pronounced effects in this sample, future versions of PEBL will allow the experimenter to select among the original (A B C D) IGT (Bechara et al., 1994) or the variant (A' B' C' D') task (Bechara, Damasio & Damasio, 2000). Due to this key methodological difference, results from the $_{PEBL}$IGT (Hawthorne, Weatherford & Tochkov, 2011; Lipnicki et al., 2009a; Lipnicki et al., 2009b) are unlikely to be identical to what would be obtained if the $_{PAR}$IGT was employed.

These datasets also provided an opportunity to identify substantial individual differences with both the $_{PAR}$IGT and the $_{PEBL}$IGT. One concern with quantifying decision making with the IGT is that there is considerable heterogeneity of responding, even by normal (i.e., neurologically intact) participants (Steingroever et al., 2013). For example, Carolselli and colleagues determined that over two-thirds (69.5% versus 68.4% in the present study) of university students completing an IGT based on Bechara et al. (1994) made more selections from Disadvantageous than Advantageous Decks (Caroselli et al., 2006). A similar pattern with the $_{PAR}$IGT was also identified in a separate sample with 70.3% of college students from the southwestern US again choosing Disadvantageous over Advantageous Decks (Piper et al., 2015b). If forced to choose whether the median participants in this college student sample (Fig. 4) show a response pattern more similar to the typical control or to a patient (EVR 318) from Bechara et al., 1994, we would select the lesioned profile. Similarly, Bechara and colleagues noted that over one-third (37%) of controls fell within the range of ventromedial prefrontal lesion group when using the ascending (A' B' C' D') paradigm (Bechara & Damasio, 2002). Findings like this, as well as the present outcomes (i.e., almost half favoring the Disadvantageous Decks with the $_{PAR}$IGT) call into question the clinical utility of this test (see also the meta-analysis by Steingroever et al., 2013). The IGT is likely measuring important elements of executive function but we are skeptical that preferential selections from Disadvantageous Decks is a specific index of a brain insult.

The benefit of open-source neurobehavioral tests like the $_{PEBL}$IGT is that the source code is readily available (see Supplemental Information) and anyone, independent of their financial resources, can use PEBL. This contributes to the democratization of science. It must also be emphasized that there is substantial room for improved construct validity and test-retest reliability for the IGT (*Buelow & Suhr, 2009*). Anyone, even with limited computer programming expertise, who is interested in modifying task parameters and generating future generations of decision making paradigms may do so, which, hopefully, will result in tests that have even better psychometric properties (e.g., the new $_{PEBL}$IGT by *Bull, Tippett & Addis, 2015*). The transparency and flexibility of PEBL are advantages over proprietary computerized neurobehavioral applications. Full disclosure of all methodological information including the underlying programming of computerized neurobehavioral tests is consistent with the dissemination policy of the *National Science Foundation (2015)* and others. However, the modifiability of PEBL is a bit of a double-edged sword in that tasks like the IGT have undergone substantial refinement over the past decade. At a minimum, investigators that make use of PEBL, PAR, or other applications must include in their methods sections the version of the software they utilized.

One potential limitation of this report is the samples consisted primarily of young adult college students, whereas in clinical settings, these tests are used across the lifespan (children to adult) with a broad range of educational and mental, and psychological profiles. However, a restriction of range for the dependent measures (see Table 2 and the range of the Minimum and Maximum on both $_{PAR}$IGT and $_{W}$DS) does not appear to be an appreciable concern for this dataset, possibly because both cohorts included some individuals with ADHD, including ones not currently taking their stimulant medications. As noted earlier, the characteristics of this convenience sample is more comparable to those employed by others (*Caroselli et al., 2006*). The PEBL software currently consists of over one-hundred tests of motor function, attention, learning, memory, and executive function in many different languages, and so additional validation studies with more diverse (age, ethnicity, socioeconomic status, computer experience) samples are warranted. Second, the sample size ($N = 44$–$47$/cohort) was sufficient to identify correlations across platforms ($r_{crit} > .20$). However, this number of participants is on the low-end to identify correlation differences (Table 3 or the 95% CI of noted correlations) between applications. Additional, and better powered, IGT psychometric investigations are needed which employ all four test sequences ($_{PAR}$IGT$_{1st}$–$_{PAR}$IGT$_{2nd}$; $_{PAR}$IGT$_{1st}$–$_{PEBL}$IGT$_{2nd}$, $_{PEBL}$IGT$_{1st}$–$_{PAR}$IGT$_{2nd}$ $_{PEBL}$IGT$_{1st}$–$_{PEBL}$IGT$_{2nd}$) for test development. Third, the $_{P}$DS was modified so that numbers were presented only via audio. These findings on the criterion validity of the $_{P}$DS with the $_{W}$DS may not be applicable to different modes (e.g., visual only, or visual and auditory) of stimuli delivery. Possibly, a fourth limitation is the few procedural differences between the $_{PAR}$IGT and $_{PEBL}$IGT (Table 1) were not identified until after the data had been collected. Identification of all the essential procedural variables for proprietary measures is not trivial, nor is that even a goal for PEBL test development. Future releases of PEBL (0.15) will however contain an IGT based on the *Bechara, Tranel & Damasio, (2000)* as well as other procedural variations (*Bull, Tippett & Addis, 2015*).

## CONCLUSIONS

This report identified a high degree of consistency between the $_C$CPT and $_P$CPT, the $_W$DS and the $_P$DS Forward, and the $_{PAR}$IGT and $_{PEBL}$IGT. Further procedural refinements in this open-source software battery will continue to enhance the utility of the PEBL to investigate individual differences in neurocognition.

## ACKNOWLEDGEMENTS

Thanks to Shawn Ell, PhD and members of the Ell lab for use of their laboratory space. Frank Barton provided technical assistance. Shelbie Wolfe and Samantha Munson assisted in data collection. Melissa A. Birkett, PhD and Peter N. Bull, MSc provided feedback on earlier versions of this manuscript.

### Funding

Commercial software for this project was provided by the Husson University School of Pharmacy, the National Institute of Environmental Health Sciences (T32 ES007060-31A1), and National Institute of Drug Abuse (L30 DA027582). The funders had no role in study design, data collection and analysis, decision to publish, or preparation of the manuscript.

### Grant Disclosures

The following grant information was disclosed by the authors:
Husson University School of Pharmacy.
National Institute of Environmental Health Sciences: T32 ES007060-31A1.
National Institute of Drug Abuse: L30 DA027582.

### Competing Interests

Shane T. Mueller is an Academic Editor for PeerJ.

### Author Contributions

- Brian Piper conceived and designed the experiments, performed the experiments, analyzed the data, prepared figures and/or tables, reviewed drafts of the paper.
- Shane T. Mueller contributed reagents/materials/analysis tools, wrote the paper, reviewed drafts of the paper.
- Sara Talebzadeh and Min Jung Ki performed the experiments, wrote the paper, reviewed drafts of the paper.

### Human Ethics

The following information was supplied relating to ethical approvals (i.e., approving body and any reference numbers):

Willamette University and Husson University approved all procedures.

### Data Availability

http://pebl.sourceforge.net/.

## Supplemental Information

Supplemental information for this article can be found online at http://dx.doi.org/10.7717/peerj.1772#supplemental-information.

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
