# Peer review of "Evaluation of the validity of the Psychology Experiment Building Language tests of vigilance, auditory memory, and decision making"

_PeerJ, doi:10.7717/peerj.1772_

## Round 0.1 · original submission · Major Revisions

Dear Brian and colleagues,

I have received two reviews of your manuscript and have been through the manuscript myself. Many of my own thoughts are mirrored in the comments from Reviewer 1. I’ll recount the major components that I feel should be considered and also include a series of specific points that relate to these and additional thoughts. As we make quite a few comments, I see that these constitute major revisions but most are aimed at clarifying what's already included. I feel that addressing these points will help to provide a more direct and higher impact article.

Best wishes,
Nic

Rationale
It would also be useful to explicitly clarify how the aims of the paper relate to each item in the results section. At times, the rationale for certain statistics would help to clarify this. It might be the case that some statistics aren’t necessary to achieve your aims.

Validity criterion
It would be helpful to outline the criterion against which acceptable validity will be judged. For correlations between measures, this is straight forward, however, for the IGT pattern/response analysis, it would be helpful to have this clearly laid out in the introduction.

IGT inclusion and analyses
Regarding the objective to validate use of PEBL, including a task that cannot be validated within-subjects as the other tasks seems counterintuitive. In addition, completing the task with both versions (PEBL & traditional) and then not analysing the repetition limits the potential scope of the work. Please consider including the correlation between repetitions in line with the analyses of the other two tasks, in addition to the pattern analysis that is reported. If nothing more, this provides evidence regarding the mentioned practice effect.

Specific: (please note, page numbers relate to the review pdf)
Page 3: Abstract, line 55/56: “DS Forward was significantly greater than DS Backward independent of the test modality.”
> consider dropping this sentence from the abstract or rephrasing to draw attention to the correspondence in estimated forwards and backwards spans (i.e., M(SD) and t-tests, rather than correlations)
> I’d also recommend include 95% confidence intervals for the correlations to give the range of all the estimates throughout
> Consider adding correlations for the IGT to be consistent with the comparisons for the other tasks. [following reading the rest of the article and I see why this wasn’t included but see below point for considering this as an inclusion]
Page 5, line 71: “20,000 times/year ”
> I think this rate would be best accompanied by the number of years that it’s been available or across which this statistic applies
Page 7, line 113: “A string of numbers are presented…”
> suggest ‘A string of numbers is presented…
Page 8, lines 155 to 157: “Although not specified a priori, the IGT dataset was also used to critically examine the sensitivity of the IGT to identify clinically meaningful individual differences in decision making abilities.”
> Without a clinical sample and/or inclusion of standardized clinical measures, it’s not clear how this can be achieved with the current data. Might be the case that more explanation is needed.
Page 9, line 170: “modified from the default in PEBL version 0.11”
> as this paper’s investigating PEBL versions of tasks, it’s important to include more detail of this modification
Page 9, line 172: “…computers running Windows and …”
> Worth reporting the operating system and computer specifications for completeness
> similarly, the operating system and specs for the Dell laptops (line 188) and touchscreens should be included
> more details are reported for the IGT laptop (line 204) – please adjust each of these computer references to cover the same information, you never know when it’s going to relevant for replications or future work
Page 11, lines 207-209: “Due to pronounced practice effects with the IGT (Bechara et al. 2000a; Birkett et al., 2015; Verdejo-Garcia et al. 2007), only data from the IGT administered first was examined. ”
> Useful to report what’s happening with the repetition here when you’re looking at validity – e.g., does one format fair better in terms of repetition
> I’d consider including the correlations between the different versions of the tasks as there should be a fair degree of overlap. You’ve got the data and I’m sure it’s something readers will be curious about so might as well be included.
Page 11, lines 218/-20: “The distribution on some measures (e.g. RT), were, as anticipated, non-normal, therefore both Pearson (rP) and Spearman rho (rS) correlation coefficients were completed. ”
> Please include the rationale for this and consider whether just reporting the Spearman statistics for the non-normal distributions would be best
Page 11, line 228: NET
> please define this abbreviation
> people quite commonly do some cleaning of reaction time data – any reason why this wasn’t applied here? Consider including justification.
Page 13, line 233: Results & correction for multiple comparisons
> There are a lot of statistical tests in here – please consider adjusting for multiple comparisons
Page 13, line 234: CPT
> Consider re-abbreviating terms like this within each section, just for readers who don’t approach the paper in a linear fashion
Page 38, Table 2
> Please include the units of measurement for reaction time
> consider reporting median reaction time in addition to or rather than mean (and inter-quartile range), can be helpful when the distributions aren’t normally distributed.
Page 13, line 238: Pearson & Spearman
> I think you’re best to just report the most appropriate correlation coefficient, dependent upon the distributions. This is especially the case regarding the point made on line 240 about removing an outlier – spearman should be more stable. This is especially the case for Omission errors (line 241)
Page 13, line 238/239 “Reaction time variability was also moderately high”
> please check this sentence – presumably this is the between version correlation for reaction time variability rather than the estimate of variability per se
> consider including a scatter plot for reaction time variability to be consistent with the other dependent variables
Page 13, line 240 to 249: Mean reaction time
> Probably makes more sense to present the descriptive statistics like the mean before the correlations.
> in addition, please indicate what the error index is for these descriptives
> please include estimates of effect size
> please consider whether the distributions (reported to be non-normal in some cases) would be better compared using non-parametric statistics.
> identifying which distributions are non-normal would be useful as well
Page 13, line 250: “An analysis of the intra-test Spearman correlations among the variables of each test was ”
> please include a rationale for these comparisons and why it was only reported for the PEBL version of the task (note: this seems counter to the validation aim of the article)
Page 14, lines 255/256: “The correlation between Forward and Backward was moderate (rP(45) = .43, p < .005; rS(45) = .41, p < .005).”
> Please include the rationale for this comparison – it doesn’t seem necessary with respect to the aims

Reviewer 1 ·

Basic reporting

The manuscript has sufficient background to justify its main objective of verifying the validity of the PEBL tests relative to their widely used non-PEBL counterparts. The methods, results and discussion comprise a coherent and methodologically useful investigation, which could be of value to researchers using the PEBL tests in their work.

The main area of concern is that the manuscript requires some clarifications to the results section, in order to make it clear how each of the analyses address the authors’ main objective of assessing the criterion validity of the PEBL measures relative to those obtained from non-PEBL tests. These concerns are described in more detail in the following sections.

Experimental design

The primary research question of the study is clearly defined. However, the secondary research aim of using the IGT data to examine the tests’ sensitivity to individual differences in decision making is not clear, and not sufficiently supported in the introduction. It is also unclear why the IGT dataset was used in this way if it was not decided a priori. The manuscript would be more coherent if the authors could provide an explicit justification for this secondary objective in the introduction, and why it is important in relation to the main study objective of verifying the validity of the PEBL tests.

Some specific questions about the experimental design are below, which relate to how the study design was decided upon. The methods section would be strengthened if the authors could clarify these points to ensure the planning that went into their design and running of the study is clear.

Page 9 line 161
Were students completing the DS/IGT reimbursed?
Was there a reason two separate cohorts were used, instead of running the four PEBL and non-PEBL tests within the same sample of participants?

Page 11 line 209
If the authors knew about practise effects with the IGT and only intended to analyse the first test administered, why did they administer both tests to each participant? Did they look at the data from the second IGT test at all?

The use of only one IGT test from each participant also means the PEBL-IGT versus par-IGT comparison was between participants, with data used from ~23 participants who did the test first. This is a substantially lower number of participants compared to the CPT and DS tests, in which the PEBL and non-PEBL versions were compared within-participants with >40 participants. Did the IGT therefore have sufficient power to detect any differences between the PEBL and non-PEBL versions?

Validity of the findings

The results of the CPT and DS tests address the study’s main objective of assessing the validity of the PEBL tests relative to their non-PEBL counterparts. However, it is not clear in the IGT results section whether the data were statistically compared between the two tests, or how the reported analyses explicitly address the validity of the PEBL-IGT relative to the par-IGT.

The main limitation of the findings concerns the need for clarification in parts of the results section, and the corresponding conclusions drawn in the Discussion. Specific concerns are listed below. The clarification of these points by the authors would greatly strengthen the conclusions they draw from their results and make it clear how the data support their conclusions.

Page 11 line 223
It would help to clarify the sentence, “Differences in correlations between the p-CPT and c-CPT were evaluated with a Fisher r to Z transformation…” – correlations between the CPT tests and what? It sounds as if the authors are comparing the correlation coefficients between each CPT test and something else but it isn’t clear what this is.

Page 11 line 226
It would be helpful here to explain the statistical tests chosen, and how they relate to the main objective of assessing the validity of the PEBL tests in relation to the non-PEBL tests. It is not clear how these analyses address this objective. Assessing RT variability, accuracy, and the correlation between each pair of tests are important for assessing whether the data obtained by the PEBL and non-PEBL version of each test is comparable but it may be helpful for the reader if these analyses were made clearer in relation to the question addressed by the study.

Page 13 line 239
The sentence “RT variability was moderately high” could be clearer – does this mean that the correlation between the degree of RT variability on each test was moderately high?

Page 13 line 245
What was the direction of the RT difference – were responses faster on the PEBL or non-PEBL test?

Page 15 line 285
Is there a reason the authors did not use a between-subjects ANOVA with the factors of Test (par-IGT vs. PEBL-IGT) by Block (Blocks 1-5) on reaction time data? This would seem a more parsimonious way to assess if RTs differed across Blocks between the two tests. Similarly, the number of Disadvantageous/Advantageous selections in each block could be submitted to a Test x Block ANOVA to assess any difference between the two tests.

Page 15 line 296
This seems to be the first point in the IGT results section where the two tests are compared. Because of the detailed reporting of the separate data from the par-IGT and PEBL-IGT preceding this, it would be helpful for the authors to provide a brief explanation of why they have reported the data from the individual tests in such detail without directly comparing them.

Further, in the introduction (line 155) the authors state the IGT dataset would also be used to examine its sensitivity in detecting individual differences in decision making. There is no clear evidence of this in the IGT results section. It would be improved if the authors clearly stated the analyses that met their main objective of comparing the data obtained by the par-IGT and PEBL-IGT, and the analyses that met their (if justified) secondary objective of assessing the sensitivity of the IGT for picking up individual differences in decision making. Without this explanation these data are difficult to clearly interpret with respect to the study goals.

Page 18 line 368
With respect to the previous comment, it is not entirely clear from the IGT results whether there were different effects between the two tests, because it is not explicit that the two tests were directly compared. Addressing this in the results section will support the claim that there were no other differences between the par-IGT and PEBL-IGT. Note that to make this claim, an analysis such as a Test x Block ANOVA should show no effects of Test on the dependent measures of interest. It is also necessary to address the issue of power here, in terms of whether no other differences between the IGT tests could be because of the reduced power of the IGT comparison compared to the other tests (because the IGT results were from a between-subjects comparison with half the number of the participants used for the other test comparisons.)

Page 19 line 375
See comment for pg 15 line 296 – it is not entirely clear how the authors reported these individual differences. Again, clarifying the IGT results section may help the reader follow this part of the discussion. As this section of the discussion stands it is not clear how it follows on from the IGT data reported.

Additional comments

The manuscript reports a methodologically valuable investigation which will be of use to researchers using the PEBL tests, and those seeking to assess the validity of their own tests relative to established counterparts. A clarification of the points above would make the manuscript more coherent. In particular, the IGT data require more robust analyses (and/or clearer reporting of the analyses conducted) for the authors to verify there is no difference between the par-IGT and PEBL-IGT.

Based on the potential merit of this work as a reference tool for other researchers, I recommend it for publication following minor revisions to 1) clearly justify the secondary study objective, 2) justify the above methodological points, and 3) clarify the reporting of the results as described above.

·

Basic reporting

The article fulfills all the criteria for basic reporting.

Experimental design

The experimental design is sound, although I am concerned that the PEBL digit span task was administered through headphones, rather than presented on screen as it normally is. The authors should state why this change was made, and what effect this could have on the findings of the study.

Validity of the findings

The authors should consider including some effect sizes to accompany the inferential statistics. Otherwise, the results and conclusions presented are justified.

Additional comments

None

---

## Round 0.2 · Minor Revisions

Dear Brian and colleagues,

Thanks for your attention to detail on the comments from myself and the reviewers in your revised manuscript. I only requested a single re-review in this latest round of review as the comments of the other reviewer were suitably addressed. Reviewer 1 raises some additional though minor points for consideration and I would be keen for these to be considered in terms of bringing further clarification to the work.

Reviewer 1 ·

Basic reporting

The clarity of the manuscript is improved from the previous version. My major previous concern was the clarity of the results section, and the authors have worked hard to address this. I still have some minor issues with the clarity of reporting, but these points are detailed below.

Experimental design

I would suggest a few further points for clarification in the Introduction to strengthen the manuscript:

Introduction
Line 83 – After the first paragraph setting up the rationale, it would be helpful to start this paragraph by explicitly stating that your goal in this paper is to compare a subset of the PEBL tests to their established counterparts. Whilst this is stated below on line 155, on first reading it takes a moment to realise this is the goal of the paper. As such, explicitly stating it early on would be valuable for the reader.

Line 98 – Would the more formal and correct citation be “Conners and colleagues”, in place of “Keith Conners, PhD”, both here and in subsequent instances?

Line 165 – The explanation of testing the PEBL and non-PEBL versions of the tests here is a clear and useful addition. However, at the end of this paragraph it is unclear whether the authors are explaining what has previously been done, or stating that this is what they intend to do in their study. Explicitly stating what method the authors are using would also be beneficial for the reader here.

Validity of the findings

The results section is clearer with respect to the previous version of the manuscript. However, some minor points do require clarification:

Results
Line 280 – It looks like the correlation between RTs on the pCPT and cCPT is not significant even with the outlier in (p = .076). A more accurate statement would therefore be that there was a moderately-sized positive correlation, but it did not reach significance.

Line 281 – Clarification is needed of what (15.9, 23.3) mean with reference to the removal of this extreme score. Further, the authors should state how this extreme score was identified – i.e. did it fall more than 1.5 standard deviations above the mean for one of the tests? It is important to be explicit about how outliers are identified and removed in the dataset.

Line 287 – Was the extreme value (if it was indeed an extreme value according to a specific criterion) retained for these intra-test Spearman correlations? It is necessary for the authors to state this clearly, as the inclusion or exclusion of outliers will be critical for readers interpreting these correlations.

Line 300 – As the correlations between test versions and variables are often described as ‘moderate’ and ‘intermediate’, it occurs to me now that it may be valuable for the authors to explain these terms at the start of the results section. For example, are there a particular range of r-values which are standardly interpreted as to a ‘moderate’ correlation? If not, it may be useful to have a brief explanation of which correlation strengths the authors will be referring to as ‘moderate’, ‘intermediate’, and so on for the purposes of this paper, simply to aid readers’ interpretation.

Line 335 – It is difficult to parse “participants made non-significantly more selections…” It may be helpful for the authors to simply describe this contrast as non-significant, or trending on significance.

Line 340 – In the IGT results section, it would be helpful to explain that the PAR and PEBL versions of the tests are first analysed separately, before being compared here. It should also be explained precisely what analysis was done here to compare the PEBL-IGT money and the PAR-IGT money across trials, and why it was conducted. This is still not sufficiently clear, and a brief explanation of this specific choice of analysis here would again be helpful for the reader.

Additional comments

The authors have worked hard to address the concerns raised about their original manuscript, with detailed and considered responses. My primary concerns about the original manuscript were with respect to the clarity of the analyses, in terms of i) the approaches used, and ii) how the analyses addressed the study objectives. In their revision the authors have mostly addressed these concerns. In particular, the explanation of the statistical analyses (line 236) is particularly useful, as are additional clarifications in the Discussion.

With the above changes the analyses should be clearer, which will aid readers in confidently making use of the authors’ work. The manuscript should provide a valuable resource for researchers intending to use the PEBL tests or conduct similar validation studies and, pending the above minor revisions, I recommend it for publication in PeerJ.

---

## Round 0.3 · accepted · Accept

Dear Brian and colleagues,

Congratulations on the quick turn-around for your last revision! You have addressed the comments of that reviewer to my satisfaction and I am pleased to be able to recommend your work for publication. Thanks for all the work you've put into this article - it will be a useful contribution and allay any concerns researchers' have about using the PEBL platform.

One very minor point that could be addressed in the final version, is the inclusion of a reference for the effect size criteria on Lines 241/242.